# The relation between Self-Control, Need for Cognition and Action Orientation in secondary school students: A conceptual replication study

Jonne Colling[1]*, Rachel Wollschläger[1], Ulrich Keller[1], Julia Grass[2], Anja Strobel[2], Franzis Preckel[3], Antoine Fischbach[1]

1 Faculty of Humanities, Luxembourg Centre for Educational Testing (LUCET), Education and Social Sciences, University of Luxembourg, Esch-Sur-Alzette, Luxembourg, 2 Department of Psychology, Personality Psychology and Assessment, Chemnitz University of Technology, Chemnitz, Germany, 3 Department of Psychology, Giftedness Research and Education, University of Trier, Trier, Germany

* joanne.colling@uni.lu

**Data Availability Statement:** The data analyzed in our study is sensible full-cohort educational data

## Abstract

Self-Control can be defined as the self-initiated effortful process that enables individuals to resist temptation impulses. It is relevant for conducting a healthy and successful life. For university students, Grass et al. found that Need for Cognition as the tendency to engage in and enjoy thinking, and Action Orientation as the flexible recruitment of control resources in cognitively demanding situations, predict Self-Control. Further, Action Orientation partially mediated the relation between Need for Cognition and Self-Control. In the present conceptual replication study, we investigated the relations between Self-Control, Need for Cognition, and Action Orientation in adolescence ($N = 892$ 9th graders) as a pivotal period for the development of self-control. We replicated the findings that Need for Cognition and Action Orientation predict Self-Control and that Action Orientation partially mediates the relation between Need for Cognition and Self-Control. In addition, Action Orientation moderates the relation between Need for Cognition and Self-Control. This result implies that in more action-oriented students Need for Cognition more strongly predicted Self-Control than in less action-oriented students. Our findings strengthen theoretical assumptions that Need for Cognition and Action Orientation are important cognitive and behavioral mechanisms that contribute to the successful exertion of Self-Control.

## Introduction

Studied in various disciplines ranging from social, clinical, and developmental psychology to sociology, criminology, and medical sciences [1], Self-Control was found to relate to numerous forms of behavior such as a reduced tendency to substance abuse (e.g., [2–5]), better saving habits (e.g., [2,6]) and interpersonal skills (e.g., [5]), which are in turn considered relevant to conduct a healthy and successful life [1]. Thereby, Self-Control is defined as the self-initiated effortful process that enables individuals to resist behavioral, emotional or attentional temptation impulses [7–9].

(used primarily for purposes of educational quality assurance) collected in the scope of the Luxembourg School Monitoring Programme "Épreuves Standardisées" (ÉpStan; www.epstan.lu), which is a joint-venture between the University of Luxembourg's LUCET (Luxembourg Centre for Educational Testing) and the Luxembourg Ministry of Education. According to the LUCET statutes, the ÉpStan longitudinal database is only to be hosted at LUCET, and the Head of the centre manages the access under supervision of a Board composed of Academic and Ministerial stakeholders alike. Requests to access the data used in our manuscript can be directed to www.epstan.lu. In order to allow nonetheless a replication of the main findings of our study (RQ 1 to 3), we have provided the latent covariance matrices for all four structural equation models in Table A6 in S2 Appendix B.

**Funding:** The author(s) received no specific funding for this work.

**Competing interests:** The authors have declared that no competing interests exist.

Enabling individuals to regulate their cognitive and behavioral responses when facing a conflict between mutually exclusive long-term goals or values and short-term situational alluring benefits [7–9], educational research has furthermore identified Self-Control as an important predictor of academic achievement at all levels of schooling (e.g., [5,10–14]; for an overview see [11]).

Although Self-Control can be considered a well-established construct in educational research, considerably less is known on the relation between Self-Control and other (motivational) constructs that broadly tap into the recruitment of cognitive resources such as Need for Cognition (NFC) and Action Orientation (AO). In a recent study, the interplay between Self-Control, NFC and AO has been analyzed in university students [15]. Basing themselves on the assumption that cognitive engagement (NFC) and resource recruitment (AO) are needed to exert successful Self-Control, Grass et al. [15] found NFC and AO to be correlated predictors of Self-Control and AO was furthermore found to partially mediate the relation between NFC and Self-Control with a remaining direct effect between the two constructs.

The present study is aiming at conceptually replicating these findings in a younger study population of $N = 892$ secondary school students (46% female, $M_{age} = 14.94$, $SD = 1.02$) by relying on large-scale data from the Luxembourg School Monitoring Programme (ÉpStan; [16]). In line with the structure of the original study [15], we are first going to present existing research focusing on the individual constructs of Self-Control, NFC and AO–and especially so in the educational setting–before providing a theoretical background for the assumed interplay between NFC and AO in their relation to Self-Control.

## Theoretical framework

### Self-Control

Self-Control relates to a broad range of positive outcomes in life that can be summarized under the five behavioral domains of (1) achievement and task performance (e.g., at school or work), (2) impulse control (e.g., in the regulation of eating or spending), (3) psychological adjustment (e.g., emotional distress, anxiety), (4) interpersonal relations (e.g., harmonious interactions, anti-social behavior) and (5) moral emotions (e.g., shame, guilt, [5]). In a meta-analysis on the relation between Self-Control and behavioral effects across different domains including a total of 102 studies ($N = 32.648$), effect sizes varied strongly across domains with the effect being strongest for school and work performance ($r = .35$; [1]).

In educational research, cumulative empirical evidence shows that Self-Control relates to academic achievement irrespective of its operationalization (e.g., course grades, standardized achievement test scores) and at all levels of schooling (for an overview see [11]). One popular measure of Self-Control in preschool is the Delay of Gratification paradigm that examines the time a child can delay immediate reward for the benefit of a more valued, but delayed one [17]. Delay of Gratification that was measured in preschool children has been related to later academic competence in adolescents [18,19].

Although the paradigm has been criticized with regard to its replicability (e.g., smaller and rarely statistically significant associations) and the Marshmallow Test measures not only Self-Control but also other essential aspects (e.g., [20]), the findings nevertheless illustrate the positive effects of self-regulatory abilities.

In primary school students, findings were similar. In longitudinal samples (with a mean age of 7.7 [$SD = 0.6$], and 8.4 years [$SD = 1.5$], respectively), Self-Control positively predicted grade point average (GPA) in Math and Chinese [21] and tests scores in reading performance [12].

In secondary school, longitudinal studies on eight-grade students (with a mean age of 13.4 in both studies, and a $SD$ of 0.7 and 0.4, respectively) provided evidence for relations of Self-

Control to different indicators of academic success such as GPA, achievement test scores, and grade change [13,22]. Results on eight-grade students (with a mean age of 13.8, *SD* = 0.5) furthermore suggest that Self-Control predicts academic achievement incrementally above previous performance and to a larger effect than IQ [22].

Similarly, in two studies focusing on undergraduate university students (with a mean age of 20.07 [*SD* = 4.99], and 20.10 years [*SD* = 4.23], respectively), individuals high in Self-Control had a better GPA than those with low levels of Self-Control [5]. In addition, Self-Control explained incremental variance in both objective (GPA) and subjective academic achievement (self-report scales) in university students (with a mean age of 22.53 years, *SD* = 3.83), even when cognitive ability was controlled for [14].

Based on previous educational research on Self-Control, it can be concluded "that more self-controlled students thrive academically at every level of formal schooling, from kindergarten through university" ([11], p. 374). With Self-Control being of a considerable importance in the academic setting, it seems important to generate a deeper understanding on how other constructs that are broadly tapping into the recruitment of cognitive resources, such as NFC and AO, might contribute to its prediction.

## Need for Cognition

NFC is a personality trait, most commonly defined as an individual's "tendency to engage in and enjoy thinking" ([23], p. 116) with individuals high in NFC showing a higher intrinsic motivation to make use of their cognitive abilities (e.g., more active search for information and a more elaborated information processing) and individuals low in NFC showing a low intrinsic motivation to engage in cognitively challenging endeavors (e.g., preference to rely on others or on cognitive heuristics to make sense of their word, [24]).

With its focus on how individuals invest their cognitive resources, NFC has gained increasing attention in educational research, where it was found to account for individual differences in cognitive and academic variables (e.g., [25–29]). Regarding cognitive variables, NFC positively relates to general intelligence and to both fluid (intelligence as process) and crystallized (intelligence as knowledge) components ([26,30]; see [31] for no significant relation with crystallized intelligence). In addition, positive relations were identified between NFC and affective-motivational constructs such as academic self-concept (e.g., [27,28,32]) and interest (e.g., [27]).

For academic achievement as outcome variable, research findings consistently identified positive associations with NFC, irrespectively of its operationalization (e.g., school grades, standardized test results) including students ranging from primary school to university (e.g., [23,27–29,31,33–35]). In tertiary education, a meta-analysis including 217 studies on psychological correlates of university students' academic performance, Richardson et al. [36] identified a positive correlation of *r* = .19 between NFC and GPA in a total sample of *N* = 1.418 students from five different studies. With regard to primary school children, moderate positive correlations with school grades were reported [31]. In secondary school students, Preckel [29] identified weak positive correlations between NFC and school grades in Mathematics, whereas correlations with grades in German, English, and Biology were found to be nonsignificant. Similarly, Luong et al. [28] found NFC to be positively related to academic achievement measured as the mean of school grades in three different school subjects (Finnish, Foreign Language, and Math).

NFC furthermore explains incremental variance in academic achievement over and above other motivational constructs such as learning-orientation, control motivation, academic self-concept or academic interest [27,28]. Moreover, NFC was found to be mainly unrelated to

student background characteristics, such as gender (e.g., [23,29,37–39]), SES (e.g., [33,35,39,40]), language, and migration background (e.g., [33,39]).

Due to positive correlations with cognitive abilities, motivational constructs, and academic achievement while being unrelated to background characteristics, NFC can be considered as an important construct in the education setting.

## Action Orientation

Kuhl [41] introduced the construct of AO (vs. State Orientation) aiming at capturing individual differences in affect-regulation (e.g., emotions, cognitions, and behaviors) during goal pursuit. AO is defined as an individual's self-motivated change-promoting tendency to flexibly recruit control resources when facing demanding situations and to pursue self-congruent goals. In contrast, State Orientation describes a change-preventing tendency that inhibits individuals to implement actions that would allow the termination of unwanted affective states [41,42]. Kuhl [43] differentiates between three dimensions of AO contrasting (1) AO subsequent to failure vs. preoccupation, (2) prospective and decision-related AO vs. hesitation, and (3) AO during performance of activities vs. volatility.

Compared to Self-Control, the relation between AO and academic achievement has gained less attention in educational research. Investigating the effect of AO on academic achievement in undergraduate students (with a mean age of 24 years, $SD$ = N/A), AO was positively related to effort which is considered a significant antecedent of academic achievement [44]. In students enrolled in a teacher education program (with a mean age of 20.69 years, $SD$ = 3.15), students high in AO received better grades than their peers low in AO [45]. In undergraduate students (with a mean age of 20.5 years, $SD$ = N/A), AO furthermore explained incremental variance in academic achievement, over and above measures such as goal orientation and cognitive ability [46].

Besides research on the relation between AO and academic achievement at university, a number of studies have been investigating the relation between AO and behavioral correlates that are of direct importance in the educational setting. In a longitudinal study, AO was positively associated with both adaptive goal setting and successful goal striving with action oriented students (with a mean age of 21 years, $SD$ = 3.6) displaying higher levels of autonomous motivation and being less likely to pursue their academic degrees due to extrinsic incentives or social pressure [47]. Regarding self-regulated learning, decision-related AO was positively related to the use of metacognitive and resource management strategies in undergraduate students (with a mean age of 20.4, $SD$ = 2.8). It was furthermore associated with positive achievement emotions (e.g., enjoyment, pride) in undergraduate students (with a mean age of 19.83 years, $SD$ = 1.63), whereas failure-related AO was either activating (e.g., anxiety) or deactivating (e.g., boredom) negative achievement emotions [48]. By being related to academic achievement and to behavioral correlates such as goal striving, self-regulated learning, and achievement emotions, AO has to be considered as important construct in educational research that is tapping into the recruitment of cognitive resources.

## NFC and Action Orientation as predictors of Self-Control

As discussed in more detail above, Self-Control is defined as the self-initiated effortful process that enables individuals to resist temptation impulses when facing a conflict between mutually exclusive long-term goals and short-term benefits (e.g., [7–9]). With regard to Self-Control in the educational context, Duckworth et al. [11] introduced the example of a situation in which a student needs to choose between studying for a Math test and spending time on social media–two mutually exclusive behavioral responses. In light of the student's long-term

academic goal of wanting to become a doctor, checking social media reflects a short-term situational benefit, which is incongruent with the long-term goal, whereas studying for a Math test is directly related to the long-term goal of wanting to become a doctor and has thus to be considered as goal congruent. The successful exertion of Self-Control results in the student's impulse to study math (e.g., to pursue the long-term academic goal) while refraining from the situational temptation of spending time on social media (for a visualization see *Fig 1* in [11], p. 375).

To identify and understand cognitive and behavioral mechanisms that may contribute to the successful exertion of Self-Control, psychological models of Self-Control have been introduced in literature (e.g., [1,8,49–51]; for an overview see [1]). With a conflict between a certain short-term situational benefit and a long-term goal being considered as starting point of a situation that requires the exertion of Self-Control, Kotabe and Hofmann [52] identified several additional steps that are required for a successful exertion including both control motivation (aspiration to control desire) and control capacity (non-motivational cognitive resources to control desire). Linked to the mechanisms described in psychological models of Self-Control, Grass et al. [15] reflected on processes relating NFC to Self-Control (e.g., information processing, control motivation) and identified AO as a construct that potentially increases the actual effort to control applied by an individual (see *Fig 1* in [15] for a visualization).

Applied to the example of the student who needs to choose between studying for a math test or checking social media, a first step towards the successful exertion of Self-Control lies in becoming aware of the existing conflict between the two mutually exclusive behavioral responses and in the cognitive appraisal of the situation, allowing the student to identify which behavioral response is congruent with the long-term academic goal (e.g., becoming a doctor). When considering that the cognitive appraisal of conflicts between mutually exclusive behavioral responses and how these responses would align with long-term goals can be rather challenging in complex real-life situations, NFC could be an underlying cognitive mechanism contributing to the successful exertion of Self-Control. Besides a conscious cognitive effort to become aware of existing conflicts, psychological models of Self-Control underline the importance of motivational processes that allow a self-motivated mobilization of control resources (see [1] for an overview). AO should therefore further facilitate the successful exertion of Self-Control once the conflict between two mutually exclusive behavioral responses has been identified because it describes the recruitment of control resources when facing (cognitively) demanding situations ([41,42]; see *Fig 1* for a visualization of the assumed relation between Self-Control, NFC, and AO).

Whereas knowledge on the interplay between Self-Control, NFC and AO, constructs that are all broadly tapping into the recruitment of cognitive resources remains scarce, NFC and AO have individually been linked to Self-Control in empirical research. With regard to the relation between NFC and Self-Control, NFC was positively correlated with Self-Control ($r$ = .30, $p < .001$) in undergraduate students (with a mean age of 21.2, $SD$ = N/A; [53]). Similarly, NFC and Self-Control showed a significant positive correlation ($r$ = .29, $p < .001$) in 10th grade

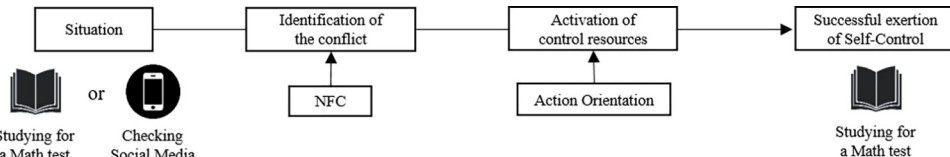

**Fig 1. The construct of Self-Control and its assumed relation to NFC and Action Orientation in the educational setting.**

students and the positive relation between NFC and academic achievement and between NFC and grade retention was mediated by Self-Control, a result indicating that the better academic performance of students high in NFC was partially mediated by higher Self-Control [54].

AO was found to be linked to Self-Control in a number of experimental studies (e.g., [55–57]). Analyzing depletion of Self-Control resources (e.g., a temporarily reduced capacity to exert Self-Control after a prior exertion), action-oriented individuals continuously allocated resources and in turn performed better on experimental tasks (e.g., critical fusion frequency test, Stroop test) than state-oriented individuals [55,56]. Whereas assumptions of Baumeister's [49] model of ego-depletion theory were supported by a meta-analysis [58], recent studies critically discuss the existence of ego-depletion (e.g., [59]) and furthermore highlight the challenges that are related to research on a multicomponent phenomenon (e.g., selection of appropriate ego-depletion tasks, lack of clear operational definitions of Self-Control; [60–62]).

In addition, AO was found to moderate the relation between general executive functioning and Self-Control in young adults (with a mean age of 22.2 years, $SD$ = 1.8; [57]). These findings underline that high general executive functioning enables Self-Control only if control resources are sufficiently mobilized or in other words if individuals are action- instead of state-oriented [57].

The interplay between the constructs of NFC, AO, and Self-Control has–to the best of our knowledge–thus far only been analyzed in a study by Grass et al. ([15], p. 3) addressing the "unresolved question to what extent Self-Control is modulated by dispositional differences in other motivation constructs". In their sample of $N$ = 1.209 university students (with a mean age of 24.43, $SD$ = 3.97), Grass et al. [15] identified NFC and AO as correlated predictors ($r$ = .32, $p$ < .001) of Self-Control and they furthermore found AO to partially mediate the relation between NFC and Self-Control with a remaining direct effect between the two constructs. These results indicate that NFC is not only associated with how individuals approach cognitive challenges but also with the actual investment of cognitive effort in situations demanding the successful exertion of Self-Control.

## The present study

Considering that both NFC and Self-Control are well-established constructs in educational research (e.g., [5,10–12,27–29,31]), it is important to generate solid knowledge on motivational mechanisms related to Self-Control by investigating how individual differences in NFC and AO relate to the prediction of Self-Control in different age groups. In this context, the present study is aiming at replicating the finding that AO partially mediates the relation between NFC and Self-Control in a younger population of 9th grade students.

In research, adolescence has been identified as a pivotal period for the development of self-regulatory capacities due to adolescent-specific neurocognitive processes (e.g., brain maturation, [63]). In addition to these biological processes, adolescence is a developmental period characterized by important cognitive and social challenges [64,65] that result in new demands on self-regulatory capacities. One of these challenges consists in the transition into secondary school, which can be considered a learning environment of special interest as it is characterized by more challenging academic structures and expectations [66]. Considering that students are faced with a higher workload [67], with an emphasis on performance goals and evaluations of achievement in competition to others [68,69], and with greater individual responsibility in managing academic requirements [66], a greater emphasis is put on secondary school students (academic) self-regulation capacities [70].

Against this background, it seems important to examine motivational constructs such as Self-Control and NFC (including their interplay with other constructs such as AO) in such a

relevant period for the development of self-regulatory capacities. The present conceptual replication study with its focus on 9[th] graders allows to generate a first understanding of whether relations between constructs reported at one level of school (e.g., university) are already established in a younger age group or whether age-specific relations can be identified that might be specific to the challenges of the respective learning environment (e.g., secondary education). Adapting the approach of a conceptual replication by testing the same hypotheses investigated by Grass et al. [15] through the use of measures adapted to the younger population of interest [71,72], we are thus focusing on the following research questions:

*a) Are NFC and AO correlated predictors of Self-Control in 9[th] graders?*

In a first step, the present study was interested in examining whether the two constructs of NFC and AO are correlated predictors of Self-Control in a sample of 9[th] grade students. Based on positive relations (of about $r = .30$) that have been identified between NFC and Self-Control (e.g., [53,54]) and on the findings of Grass et al. [15] in their sample of university students, we expect a small positive association between the two constructs. As AO was also positively linked to Self-Control in experimental studies (e.g., [55–57]), we furthermore expect a small positive relation between AO and Self-Control. Regarding the relation between NFC and AO, it can theoretically be assumed that both NFC and AO are associated with how much cognitive effort an individual is willing to invest leading to a small to moderate association. The results by Grass et al. [15] confirmed that assumption. In line with their findings in university students, we expect NFC and AO to be positively correlated predictors of Self-Control in 9[th] grade students.

*b) Does AO moderate the relation between NFC and Self-Control?*

As discussed in more detail above, psychological models of Self-Control (e.g., [1,8,49–51], for an overview see [1]) refer to underlying cognitive and behavioral mechanisms that contribute to the successful exertion of Self-Control. Whereas NFC might facilitate the first step towards the successful exertion of Self-Control by enabling individuals to become aware of an existing conflict between two mutually exclusive behavioral responses (e.g., studying for a test vs. checking social media) through a cognitive appraisal of the situation, AO allows the self-motivated recruitment of the necessary control resources to pursue the behavioral response that is congruent with the individual's long-term goal (e.g., becoming a doctor).

We therefore investigated whether AO moderates the relation between NFC and Self-Control. A moderator variable affects the strength of the relation between two other variables (e.g., NFC and Self-Control) in such a way that this strength differs depending on the level of the moderator (e.g., AO, [73,74]). Applied to the constructs of interest in the present study, lower levels of AO might result in less recruitment of cognitive resources and a reduced motivation to invest cognitive effort even in individuals high in NFC (i.e., weaker relation between NFC and Self-Control). Higher levels of AO, on the other hand, could foster the actual investment of cognitive effort (i.e., stronger relation between NFC and Self-Control). Thus, we analyzed whether AO moderates the relation between NFC and Self-Control in 9[th] graders or in other words whether the relation between NFC and Self-Control remains stable across different levels of AO. Considering that the relation between NFC and Self-Control was identified to be stable across different levels of AO in university students, we do not expect AO to change the relation's strength or direction in secondary school students.

*c) Does AO mediate the relation between NFC and Self-Control?*

Besides having an impact on the strength of the relation between two variables (moderation), the same variable can also mediate the relation between the two variables in question

(e.g., [74], see [75] for a detailed explanatory example). A mediator variable is defined as the intermediary process or mechanism through which one variable relates to another variable, or in other words, it allows to understand the relation more completely by assessing the extent to which the relation between two variables (e.g., NFC and Self-Control) is direct or indirect via a mediator (e.g., AO, [73,74,76]).

Following this line of thought with regard to our investigation, it can thus be expected that AO partially mediates the relation between NFC and Self-Control besides moderating it as we assume additional processes linking NFC to Self-Control that go beyond the recruitment of control resources (e.g., elaborated information processing, higher motivation to approach cognitively challenging situations; see *Fig 1* in [15] for a visualization). With regard to the finding by Grass et al. [15] that AO partially mediates the relation between NFC and Self-Control with a remaining direct effect between the two constructs in university students, we expect to confirm this partial mediation hypothesis in secondary school students.

## Methodology

### Procedure and participants

The Luxembourg School Monitoring Programme (ÉpStan, [16]) offers a standardized record of academic achievement by analysing at the beginning of each learning cycle whether the expected educational goals of the previous learning cycle have been achieved in various key school areas (e.g., German, French and Mathematics) at primary and secondary school. Being a collaboration between the Luxembourg Centre for Educational Testing and the Ministry of Education, Children and Youth, the ÉpStan have a legal basis and approval from the National Commission for Data Protection. Appropriate ethical standards were respected [77] and participating children, their parents or legal guardians were duly informed and had the possibility to opt-out. To ensure accordance with the European Data Protection Regulation, all analyses were conducted with a cross-sectional anonymized dataset of a full cohort of 9th graders ($N$ = 5.814) allocated to the academic or intermediary school track in secondary schools in Luxembourg.

Besides measures on academic achievement, the ÉpStan furthermore include an encompassing student questionnaire assessing personality, motivational and social measures that are of relevance in the educational setting (e.g., academic interest, academic self-concept, school and class climate). As the student questionnaire in the ÉpStan is presented to participants in both German and French, three different people translated the scales that were not available in these languages from their original versions (see *Measures* for more details). After a comparison of the three individual translations, potential discrepancies were discussed until a common translation was found. A native speaker subsequently validated the final translation of each scale. The questionnaire is presented computer-based. Students completed it at school in the presence of a teacher after having taken the academic achievement tests. All answers were confidential and directly stored on a dedicated platform without requiring a manual encoding.

The student questionnaire including the measures that are of relevance for the present study was presented to a subsample of $N$ = 1.678 students with the NFC scale presented first, followed by the Self-Control scales and the AO scale, respectively. Out of this subsample, all students that answered at least 70% of the items on each scale assessing our main variables of interest were retained for the analyses (see *Data Analysis* for more details). Following the logic of a conceptual replication study [71,72], age-appropriate scales were used to assess each construct (see *Measures* for more details). The final sample entails $N$ = 892 students (46% female, 54% male, $M_{age}$ = 14.94, $SD_{age}$ = 1.02) and a comparison to the 9th grade students allocated to the highest or intermediary school track that were not presented with the measures of interest

($N$ = 4.136) indicates that the subsample can be considered as representative in aspects such as student background variables (e.g., gender, SES, language and migration background) and academic achievement with the overlap being slightly more aligned with the profile of students attending the highest track (see *S1 Appendix* for more details).

## Measures

**Self-Control.**   In line with the original study [15], the complexity of Self-Control as a construct has been taken into consideration by assessing both Trait Self-Control and Effortful Control. Trait Self-Control was assessed by the German [78] and the French [79] versions of the 13-item Brief Self-Control Scale [5], which is specifically designed to capture an individual's self-perceived capacity to exert effortful control over behavioral responses in the pursuit of long-term goals. Effortful Control was assessed using a shortened 12-item version of the Early Adolescent Temperament Questionnaire-Revised [80] that was translated into German and French (see *Participants and procedure* for details on the translation procedure). The Early Adolescent Temperament Questionnaire captures individual differences in the exertion of executive control in everyday life such as the capacity to focus and shift attention as desired (attentional control), the capacity to inhibit inappropriate behavior (inhibitory control) and the capacity to perform a task when there is a strong tendency to avoid doing so (activation-related control). Items of all Self-Control scales were presented to the students on a 4-point Likert scale ranging from "*not true*" to "*true*".

**Need for Cognition.**   NFC was assessed using the 14-item NFC-KIDS scale [39] in its original German version and in a French translation. The psychometric quality of the NFC-KIDS scale has been assessed and validated in various studies with both primary and secondary school students [27,28,39]. It consists of 14 short and positively worded items that are adapted to primary and secondary school students with regard to linguistic complexity (e.g., "*I like to work on problems that require a lot of thinking*") and to the age-related context they are living in (e.g., "*In school, I want to understand everything exactly*"). Item answers were given on a 4-point Likert scale ranging from "*not true*" to "*true*".

**Action Orientation.**   Conforming to the original study [15], AO was assessed by the German [43] and French [81] versions of the Action Control Scale [43]. For the research interest of the present study, the two subscales of preoccupation (AOF) and hesitation (AOD) were measured with 12 dichotomous items each. Items of both subscales were presented alternately. The Action Control Scale confronts participants with a situation in each item (e.g., "*When I know I must finish something soon, . . .*") in which they have to select one out of two potential reactions that rather applies to them (e.g., "*I have to push myself hard to get started*" or "*I find it easy to get it done and over with*"). Answers reflecting low AO were coded 0, while answers reflecting high AO were coded 1.

## Data analysis

Descriptive statistics and reliabilities (McDonald's ω) were computed for NFC, Self-Control, and AO. In line with the manual guidelines for the NFC-KIDS scale [39], a minimum of 10 out of 14 NFC items had to be answered for the measure to be analyzable. It is in this context, that we only included participants that answered at least 70% of the items on each scale of interest (NFC, Self-Control and AO) in our final sample of $N$ = 892 students. For students with no answers on up to 30% of the items of each scale, the mean score of the answered items on the total scale replaced missing values before sum scores were created for each measure. Although replacing missing values by the mean score of the answered items on the total scale is an established method when it comes to handling missing data (e.g., [82]), it has some

important shortcomings such as the underestimation of standard errors, the loss of information on the relationship between variables and the artificial deflation of a variable's variance that need to be taken into consideration when interpreting the results [83–85]. Nevertheless, this procedure respects the manual guidelines for treating missing data in the NFC-KIDS scale [39] and as no clear indications were found for the other measures of the present study, the 70% rule was subsequently applied to the scales of Self-Control and AO.

In line with the data analysis approach of the original study [15], Structural Equation Modeling was used as this latent variable approach allows to analyze relations between NFC, Self-Control, and AO on a construct level while controlling for measurement errors. As stated by Grass et al. [15], their research interest was not to examine the factor structure of the three individual constructs but to understand the interplay between NFC, Self-Control, and AO at the construct level. Therefore, they parceled all manifest indicators into item parcels using an item-to-construct balance parceling technique (for NFC and Self-Control) and a combination of an item-to-construct balance parceling technique and a domain-representative parceling technique (for AO) following the recommendations of Little et al. [86]. Following this procedure, separate principal component analyses with a one-factor solution for all items of each instrument assessing NFC and Self-Control were performed. For AO, a two-step procedure was applied where factor loadings were calculated and ranked separately for each dimension before allocating the items to parcels in line with their respective factor loadings including three preoccupation and three hesitation items each. Items were allocated to three or four parcels (based on the total number of items respectively) and sum scores were created for each identified parcel.

In order to answer the guiding research question of whether AO moderates or mediates the relation between NFC and Self-Control in our sample of 9th grade students, we applied the same procedure as in the original study and tested a baseline model (A), a moderation model (B), a mediation model (C), and a complete mediation model (D) before comparing model fits with each other (for a visualization of the models see *Fig 3*). The moderation model (B) entails a latent interaction variable of NFC and AO that was computed using orthogonalized product indicators. All possible product terms of the predictor indicators were calculated in a first step before regressing each product on all indicators of NFC and AO. The resulting residuals of these regressions were, in turn, used as indicators of the latent interaction variable in the moderation model (B).

The nested data structure was taken into account by the ANALYSIS = COMPLEX function in which class membership was the cluster variable. The robust $x^2$ statistic, the root mean square residual (RMSEA), the comparative fit index (CFI), the Tucker Lewis Index (TLI), and the standardized root mean square residual (SRMR) were used in order to interpret model fit. In line with the original study both incremented fit indices comparing hypothesized models to a baseline model assuming independence of all variables and residual-based fit indices to evaluate the amount of error of a model estimation were included and a model was considered to have an acceptable fit with RMSEA < .07, SRMR < .10, CFI > .93 and TLI > .90. Bayesian Information Criterion (BIC) was used in addition to the other fit indices in order to compare the partial mediation model (C) to the complete mediation model (D; [87]).

## Transparency and openness

In the present study, data were analyzed using R Version 2.5 (e.g., descriptives) and SPSS Version 25 (e.g., creation of orthogonalized product indicators). For Structural Equation Modeling, Mplus Version 8 [88] was used. The section *Participants and Procedure* describes the handling of missing data (for NFC, Self-Control and AO) in detail and gives further

information on the determination of our analytic sample. The data analyzed in this study is sensible educational policy data from the Luxembourg School Monitoring Programme and has been kindly made available for the present study's specific secondary analysis. Whereas it cannot be publicly shared, requests to access the data can be directed to www.epstan.lu. Inter-correlations between items parcels and latent variable covariance matrices for all four models can be found in S2 Appendix. The study was not preregistered.

## Results

### Descriptive statistics and reliabilities

Descriptive statistics for all the manifest NFC, Self-Control, and AO scales and for each parcel can be found in *Table 3A* of S2 Appendix. Reliabilities measured by McDonald's ω at parcel level were ranging from acceptable to high with .92 for NFC, .74 for Trait Self-Control, .75 for Effortful Control, and .70 for AO. Parcels created based on principal component analyses consisted of three (NFC) to six items (AO) each. Average factor loadings of items on the respective factor were calculated for each parcel and the average loadings were between .70 and .71 for NFC, between .46 and .47 for Trait Self-Control, between .31 and .32 for Effortful Control, and between .39 and .43 for AO.

### Preliminary analyses

In line with the original study [15], the measurement model of Self-Control has been determined before computing the baseline, moderation and (partial) mediation models. A one-factor model assuming that all indicators of both Trait Self-Control and Effortful Control load on one common factor was compared to a second-order model. By including two first-order factors corresponding to Trait Self-Control and Effortful Control respectively and one second-order factor reflecting their shared variance, the structure of this theoretically preferred (see *Measures* for details) second-order model took "into account that while both instruments focus on different behavioral aspects of Self-Control, they nevertheless assess a common construct" ([15], p. 10). In the second-order factor model, the factor loadings of the two first-order factors (Trait Self-Control and Effortful Control) on the second-order factor were set equal due to a high latent correlation of .80 when replacing the second-order factor by a latent correlation between the two latent first-order factors. Results for the two measurement models of Self-Control can be found in Table 1.

When looking at the $x^2$ values of the two measurement models, results indicated that the second-order model ($p = .006$) fitted the data better than the one-factor model ($p < .001$). Due to the studies large sample size, $x^2$ values were likely to be statistically significant and other fit indices were thus taken into account. Whereas the one-factor model did not yield a good fit to the data (e.g., RMSEA > .07), the second-order model met all cut-off rules for RMSEA, SRMR,

**Table 1. Fit indices for the measurement models of Self-Control in 9th grade students.**

| Model | $x^2$ | df | P | RMSEA [90% CI] | SRMR | CFI | TLI |
|---|---|---|---|---|---|---|---|
| One-factor model | 90.728 | 9 | < .001 | .101 [.083, .120] ** | .039 | .948 | .914 |
| Second-order model[a] | 21.654 | 8 | .006 | .044 [.022, .066][ns] | .019 | .991 | .984 |

*Note*. N = 892. All models were estimated with maximum likelihood estimation (MLR) with robust standard errors. CI = Confidence Interval.

[a] Unstandardized loadings of the first-order factors on second-order factor were set equal. $\Delta x^2_{scaled} = 95.194$, $\Delta df = 1$, $p < .001$.

[ns]nonsignificant.

**$p \leq .01$.

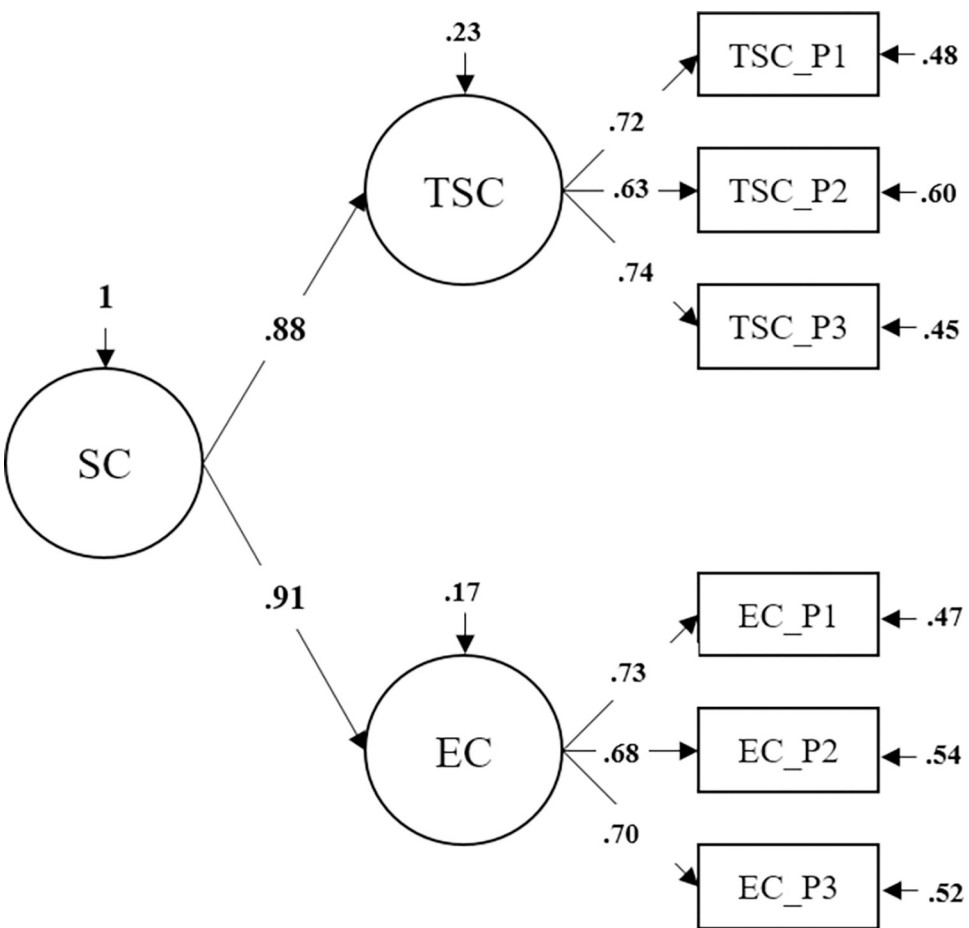

**Fig 2. Second-order measurement model of Self-Control in 9ᵗʰ grade students.** *Note.* N = 892. SC = General Self-Control. TSC = Trait Self-Control. EC = Effortful Control. Model was estimated with maximum likelihood estimation (MLR) with robust standard errors. Item parcels were used as manifest indicator variables. Unstandardized loadings of first-order factors were set equal. All paths were significant with $p < .001$.

CFI, and TLI. The superiority of the second-order model was furthermore demonstrated by the significant $x^2$ difference test (see Note of *Table 1*). *Fig 2* visualizes the second-order measurement model of Self-Control that has been retained for all subsequent analyses.

### Results for the research questions

*a) Are NFC and AO correlated predictors of Self-Control in 9ᵗʰ graders?*

In a first step, a baseline model has been tested in which NFC and AO predicted Self-Control (for a visualization see *Fig 3A*). Model fit was acceptable with $x^2 = 236.182$ ($df = 73$, $p < .001$), RMSEA = .050 (90% Confidence Interval: [.043; .057], $p = .481$), CFI = .966, TLI = .958 and SRMR = .040. Results showed that Self-Control was predicted by NFC with β = .27 ($p < .001$) and by AO with β = .50 ($p < .001$). The model furthermore identified NFC and AO as correlated predictors of Self-Control ($r = .23$, p < .001).

*b) Does AO moderate the relation between NFC and Self-Control?*

After establishing the baseline model, we were interested in understanding whether AO moderates the relation between NFC and Self-Control in 9ᵗʰ graders. Therefore, a latent interaction

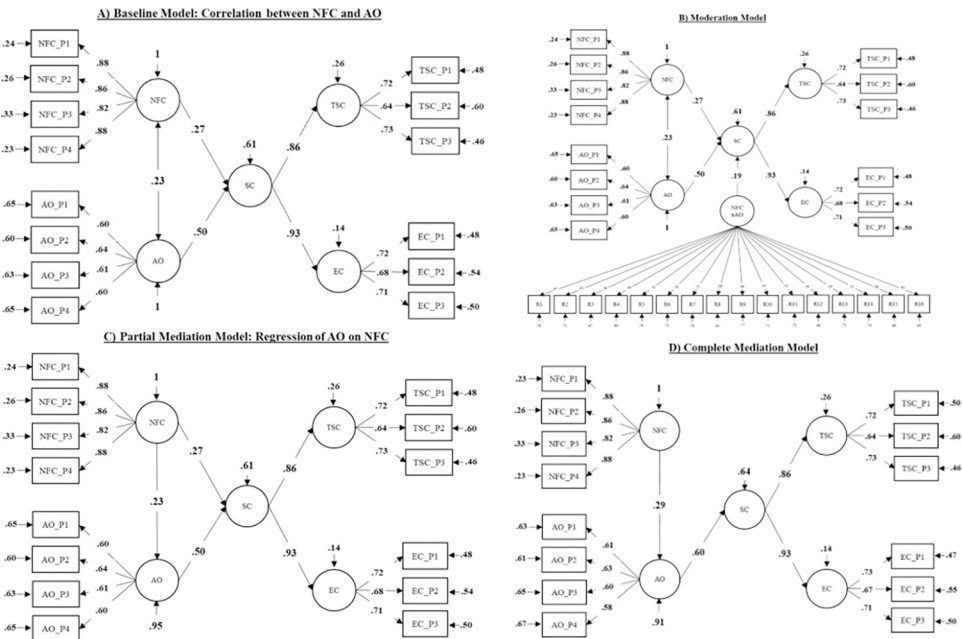

**Fig 3. Structural equation models of NFC and AO predicting Self-Control in 9th grade students.** *Note.* N = 892.
SC = General Self-Control. TSC = Trait Self-Control. EC = Effortful Control. AO = Action Orientation. Standardized
estimates displayed. Models estimated with maximum likelihood estimation (MLR) with robust standard errors. Item
parcels were used as manifest indicator variables. (A) Baseline Model. (B) Moderation Model. Interaction term
between NFC and AO (NFCxAO) calculated with residual indicators. Intercorrelations between residuals were allowed
but are not displayed. (C) Partial Mediation Model. (D) Complete Mediation Model. Baseline Model (A) similar to
Partial Mediation Model (C) except for the relation between NFC and AO being bidirectional (correlation) in (A) and
unidirectional (regression) in (C).

between NFC and AO has been added to the baseline model (see *Data analysis* for details on the
computation of the latent interaction). Model fit was excellent with $x^2$ = 333.644 ($df$ = 352, $p$ =
.751), RMSEA = < .001 (90% Confidence Interval: [.000; .009], $p$ = >.999), CFI = 1.000,
TLI = 1.000 and SRMR = .025. Results of the moderation model (*Fig 3B*) allowed the identification
of a significant interaction between NFC and AO (NFCxAO) with β = .19 ($p$ < .001) suggesting
that AO moderates the relation between NFC and Self-Control (β = .27, $p$ < .001).

*c) Does AO mediate the relation between NFC and Self-Control?*

In order to test whether AO partially mediates the relation between NFC and Self-Control,
the correlation between NFC and AO set in the baseline model (A) was transferred into a
regression of AO on NFC (*Fig 3C*). Model fit was acceptable with $x^2$ = 236.182 ($df$ = 73, $p$ <
.001), RMSEA = .050 (90% Confidence Interval: [.043; .057], $p$ = .481), CFI = .966, TLI = .958
and SRMR = .040. The total relation between NFC and Self-Control was β = .39 ($p$ < .001)
with a stronger direct (β$_{dir}$ = .27, $p$ < .001) than indirect relation through AO (β$_{ind}$ = .12, $p$ <
.001). Wanting to understand whether AO fully or partially mediates the relation between
NFC and Self-Control, this model has been compared to a complete mediation model in
which the direct path of NFC on Self-Control was fixed at 0 (*Fig 3D*). Model fit was acceptable
with $x^2$ = 281.456 ($df$ = 74, $p$ < .001), RMSEA = .056 (90% Confidence Interval: [.049; .063], $p$
= .073), CFI = .957, TLI = .947 and SRMR = .065. Based on the significant $x^2$ difference test
($\Delta x^2_{scaled}$ = 33.508, $\Delta df$ = 1, $p$ < .001), the partial mediation model (C) with a remaining direct
relation between NFC and Self-Control was superior when compared to the complete media-
tion model (D). All fit indices for the four models can be found in *Table 2*.

**Table 2. Fit indices for the baseline, moderation, partial and complete mediation models in 9th grade students.**

| Model | $x^2$ | $df$ | $P$ | RMSEA [90% CI] | SRMR | CFI | TLI | BIC |
|---|---|---|---|---|---|---|---|---|
| Baseline (A) | 236.182 | 73 | < .001 | .050 [.043, .057] [ns] | .040 | .966 | .958 | - |
| Moderation (B) | 333.644 | 35 | .751 | .000 [.000, .009] [ns] | .025 | 1.000 | 1.000 | - |
| Partial mediation (C) | 236.182 | 73 | < .001 | .050 [.043, .057] [ns] | .040 | .966 | .958 | 47706.273 |
| Complete mediation (D) | 281.456 | 74 | < .001 | .056 [.049, .063] [ns] | .065 | .957 | .947 | 47753.346 |

*Note*. N = 892. All models were estimated with maximum likelihood estimation (MLR) with robust standard errors. CI = Confidence Interval.

[ns] nonsignificant.

## Discussion

Numerous studies have demonstrated that Self-Control is significantly related to academic achievement with more self-controlled students performing better at all levels of formal schooling (e.g., [5,10–14]; see [11] for an overview). Psychological models of Self-Control refer to underlying cognitive and behavioral mechanisms that contribute to the successful exertion of Self-Control (e.g., [1,8,49–51]; see [1] for an overview). They can be considered a theoretical framework guiding research on relevant processes behind successful Self-Control to improve understanding of this complex construct and to identify interindividual differences related to Self-Control failure versus success. By broadly tapping into the recruitment of cognitive resources, the constructs of NFC and AO could be such underlying mechanisms with NFC enabling individuals to cognitively appraise a conflict between mutually exclusive behavioral responses and to evaluate how these responses align with long-term goals. Besides more elaborated information processing, individuals high in NFC are assumed to show a higher motivation for approaching a cognitively challenging situation and they are furthermore more likely to actually activate the needed control resources, which partially relates NFC to AO allowing individuals to flexibly recruit the necessary control resources to successfully exert Self-Control once the conflict between mutually exclusive behavioral responses has been identified (for a visualization of the assumed relation between these three constructs see *Fig 1*).

Whereas both NFC and AO are positively related to Self-Control in empirical research (e.g., [53–57]), the interplay between Self-Control, NFC, and AO has–to the best of our knowledge–thus far only been analyzed in the scope of a recent study focusing on university students [15]. Tacking up on the observation that future studies on Self-Control should focus on the identification of potential moderators and/or mediators both in the academic setting in general and in various age groups [11] and aiming at conceptually replicating the findings by Grass et al. [15], the present study investigated the role of individual differences in NFC and AO in the prediction of Self-Control in a younger sample of 9th grade students while using measures adapted to the younger population of interest [71,72]. The main objectives were to analyze whether (a) NFC and AO identify as correlated predictors of Self-Control and to understand whether (b) AO moderates or (c) mediates the relation between NFC and Self-Control. In line with the results in university students, Self-Control was predicted by NFC and AO. The model furthermore identified both constructs as correlated predictors of Self-Control. In addition, AO partially mediated the relation between NFC and Self-Control with the direct relation being stronger than the indirect one through AO. Compared to a full mediation model, the partial mediation model displayed the better model fit. Whereas no significant interaction effect has been identified in a sample of N = 1.209 university students [15], results from the present study identified a significant interaction effect between NFC and AO suggesting that AO is moderating the relation between NFC and Self-Control. These findings imply that the relation between NFC and Self-Control is stronger in individuals with higher levels of AO.

## Discussion of the results for the research questions

**NFC and AO as correlated predictors of Self-Control.** In line with our expectations and with results based on university students from the original study [15], findings from our baseline model identified AO and NFC as correlated predictors of Self-Control and underlined that Self-Control was significantly related to AO ($\beta$ = .50) and to a lesser degree to NFC ($\beta$ = .27) in our sample of 9th grade students. These results add to previous empirical research in which NFC and AO were individually linked to Self-Control (e.g., [53–57]) and furthermore extend the scarce knowledge on the relation between NFC and AO by identifying them as being positively correlated ($r$ = .23). As discussed by Grass et al. [15], a small positive relation between NFC and AO can be expected considering that AO is theoretically related to the allocation of (cognitive) control resources when approaching cognitively challenging situations and that NFC correlates with expanding cognitive effort in demanding situations (e.g., [89]). Our findings are in line with this expectation and thus underline that NFC and AO as constructs both seem to tap into approaching and engaging in cognitively demanding tasks.

**AO as a mediator of the relation between NFC and Self-Control.** A mediation analysis allows to understand the relation between two variables more completely by assessing the extent to which the relation is affected by an intermediary process–the mediator variable [73,74,76]. Results from our partial mediation model revealed AO to partially mediate the relation between NFC and Self-Control with a remaining direct relation between the two constructs. While the total relation between NFC and Self-Control ($\beta$ = .39) seems to be comparable to the relation identified in university students and adds to medium associations of around $r$ = .30 reported in previous empirical research [53,54], the direct relation between NFC and Self-Control was stronger in 9th grade students than in university students for whom the direct and indirect relation between NFC and Self-Control was comparable in size. In line with our expectations and the findings in the original study by Grass et al. [15], the partial mediation model displayed a better model fit than the complete mediation model underlining that NFC and AO are both relevant for the prediction of Self-Control in 9th grade students and (partially) refer to different processes preceding Self-Control. With a remaining direct relation ($\beta_{dir}$ = .27) between NFC and Self-Control approaching a medium effect size, our findings strengthen theoretical assumptions that both NFC and AO are underlying cognitive and behavioral mechanisms, which may contribute to a successful exertion of Self-Control (for an overview see [1]), and furthermore align with the conclusion that "Self-Control depends not only on dispositions that refer to behavior very close to control processes (AO) but also on dispositions that are more broadly related to cognitive processes, such as NFC" ([15], p. 14).

**AO as a moderator of the relation between NFC and Self-Control.** Besides mediating the relation between two variables, the same variable can also moderate the relation between the two variables in question and affect its strength depending on the level of the moderator variable [74–76]. Following the theoretical reasoning that lower levels of AO may potentially reduce the actual control effort exerted once a conflict between two mutually exclusive behavioral responses has been identified, the present study has therefore in a final step analyzed whether AO moderates the identified positive relation between NFC and Self-Control in 9th graders. Considering that no significant moderating effect was reported in university students [15], we did not expect AO to be moderating the positive relation between NFC and Self-Control in secondary school students. Results from a moderation model however allowed the identification of a significant interaction of small effect size between NFC and AO ($\beta$ = .19) implying that the prediction of Self-Control by NFC is stronger in action-oriented students whereas it was found to be stable across different levels of AO in university students.

A potential explanation for this divergent result in secondary school students might lie in differences between the two samples. In the study by Grass et al. [15], their questionnaire was distributed via a university mailing list and the educational level of participants was high with 99% holding a university entrance diploma, resulting in a very homogenous sample. The sample of secondary school students assessed in the present study can in contrast be assumed to be considerably more heterogeneous regarding its educational level. As described above (see *Participants and procedure* for details), students were attending either the highest (preparing students for academic studies) or intermediary track (preparing students for professional life or further academic studies) of the Luxembourgish education system. Regarding school tracks in Luxembourg, previous national and international reports have repeatedly demonstrated that the three school tracks considerably differ when looking at their students' academic achievement with adjacent school tracks indicating performance differences of two to three years (e.g., [16,90,91]). Additionally, Grade 9 marks the final year of lower secondary education and all students are taught together within their respective tracks before being allocated to different levels of vocational training in the intermediary track or to various academic sections in the highest track based on their academic abilities and interests [92], resulting in a broader sample composition in comparison to a more homogenous groups of students attending university. With differences in the mean age of university (24.43, $SD$ = 3.97) and secondary school students (14.94, $SD$ = 1.02), a further explanation for the observed difference regarding the interaction between NFC and AO could potentially lie in an age-specific relation between the two constructs, which might be reflected in the stronger correlation found in older students ($r$ = .32 in contrast to .23).

Besides differences in sample composition, it can be assumed that the learning environment of early secondary school differs from the one in late secondary school and in university. Such differences refer to the opportunity to select classes in line with students' own academic interests, the need to apply self-regulated learning strategies, a higher personal relevance of course content, a greater individual responsibility when it comes to the management of academic requirements or the encounter of cognitively demanding academic challenges that require students to acquire more abstract and diversified knowledge (e.g., [66,93–96]).

In the light of the finding that higher levels of AO are leading to a stronger relation between NFC and Self-Control in secondary school students whereas it was stable across levels of AO in university students, it can be assumed that a student's self-motivated tendency to flexibly recruit control resources when facing (cognitively) demanding situations [41,42] comes more strongly into play in the educational environment of secondary education. This environment differs from late secondary school and from university with long-term academic goals still being less clear and charted [94,95]. Higher levels of AO in early secondary school students are thus potentially allowing them to pursue a behavior that is in line with their (still less clear) academic long-term goal (e.g., becoming a doctor) instead of giving into a short-term situational temptation (e.g., checking social media). In an education setting in which students selected a field of study that more directly aligns with their own academic interests and in turn with their long-term academic goals, it seems on the other hand that individual differences in NFC enabling individuals to become aware of an existing conflict between two mutually exclusive behavioral responses relate to Self-Control with the same strength irrespective of a student's level of AO.

Taken together, the results of the present conceptual replication study highlight that AO and NFC both explain differences in Self-Control. This supports the assumption that taking individual differences in personality into account considerably adds to our understanding of a complex construct such as Self-Control. Although AO displayed a higher predictive value for Self-Control than NFC, our findings underline the relevance of an individual's "tendency to

engage in and enjoy thinking" ([23], p. 116) as NFC predicts Self-Control partially but not exclusively through AO. They strengthen the empirical evidence on the interplay between the three constructs of Self-Control, NFC and AO by adding to the conclusion drawn by Grass et al. ([15], p. 15) that "the partial mediation [displays] indirect evidence for different psychological components and processes contributing to how individuals manage behavioral conflicts that demand Self-Control". They further extend it to a younger population of secondary school students and provide evidence for a stronger relation between NFC and Self-Control in more action-oriented adolescents. That finding implicitly underlines the importance of behaviorally implementing self-control intentions for successful Self-Control [51,52].

**Discussion of the structure of Self-Control measures.** In line with the original study by Grass et al. [15], the complexity of Self-Control as a construct consisting of various components that go beyond the inhibition of temptation impulses [8,52] was taken into account by operationalizing Self-Control by the means of different questionnaires [97]. Although the questionnaires (Trait Self-Control and Effortful Control) are derived from different theoretical backgrounds, they both assess dispositional Self-Control and are assumed to share variance on a higher level (second-order factor) and differences in behaviors related to Self-Control on a lower level (two first-order factors; see *Fig 2* for a visualization). To confirm this assumption, we compared a one-factor model assuming all indicators of Trait Self-Control and Effortful Control to load on a common factor to a second-order model reflecting the assumed shared variance of two lower-order factors. The second-order model identified as superior to the one-factor model via different fit indices. These findings are in line with the hierarchical structure of Self-Control identified in university students, and it can be concluded that the second-order factor of Self-Control is also accounting for a large amount of shared variance in our younger population of 9th graders. These results extend the findings by Grass et al. [15] and strength previous research on the convergence of different Self-Control measures, which underlined that self-report measures of Self-Control tend to share a large amount of variance whereas performance tasks (e.g., Delay of Gratification tasks, executive function tasks) seem to be assessing more specific Self-Control processes [97,98].

## Limitations and perspectives for future research

The present study took the complexity of Self-Control as a construct consisting of various components going beyond the inhibition of impulses [8,52] into account by operationalizing Self-Control by the means of two different questionnaires, which were aiming at bringing together different theoretical approaches and behavioral aspects of Self-Control [5,80]. Although previous evidence on Self-Control measures shows that self-report questionnaires tend to share a large amount of variance, performance tasks designed to measure Self-Control (e.g., Delay of Gratification tasks) seem to show low convergent validity [97,98]. It is in this context that a first limitation of the present study can be identified considering that we solely relied on self-report questionnaires to operationalize Self-Control. Our data was collected in the scope of the Luxembourg School Monitoring Programme (ÉpStan, [16]), which is assessing academic achievement (e.g., German, French and math) and personality, motivational, and social measures relevant in the educational setting (e.g., academic interest, academic self-concept, school and class climate) in full cohorts of students at primary and secondary school each year. Consequently, all variables of interest were measured by the means of an extended student questionnaire, an approach that is line with a recommendation by Duckworth and Kern [97] when researchers are facing time or budget constraints. In the context of the full-cohort educational assessment with large sample sizes and logistical boundaries, the implementation of performance tasks designed to measure Self-Control was not possible. In order to validate

the identified interplay of NFC and AO in the prediction of Self-Control, future studies should rely on a mixture of different types of measures to examine the generalizability of our results when it comes to the behavioral assessment of Self-Control.

A second limitation of the present study is that our participants attended either the highest or intermediary school track of the Luxembourgish educational system. Students allocated to the lowest track designed to prepare students that had not acquired sufficient skills in primary schools to join the intermediary track or for starting a vocational training were thus not included in our sample of 9th grade students. Considering that previous national and international reports have repeatedly found the three school tracks to considerably differ when looking at students' academic achievement with adjacent school tracks showing performance differences of two to three years [16,90,91], including lower track students would further broaden the heterogeneous population of 9th graders. It is in this context that future studies should include students irrespective of their track allocation as the activation of control resources (AO) could potentially even be of higher importance in students with lower academic achievement, resulting in a changed interplay between the three constructs of Self-Control, NFC and AO.

Considering that both the original study [15] and this conceptual replication were interested in understanding the interplay between Self-Control, NFC and AO at the construct level, all manifest indicators were grouped into parcels resulting in the fact that no observations can be made with regard to the factor structure of the three individual constructs in our sample of 9th graders. Whereas the factor structure of the NFC-KIDS scale has repeatedly been investigated in both primary and secondary school students [27–29,39], futures studies should generate solid knowledge on factor structure and measurement invariance of Self-Control and AO, allowing in turn to understand how subscales (e.g., preoccupation or hesitation in the case of AO) may affect the interplay of different constructs at the second-order level.

A further limitation is that the data of our study is cross-sectional and this design does not allow to draw conclusions on causality of the identified relationships between Self-Control, NFC and AO. In the light of initial evidence for reciprocal relations between Self-Control and NFC [99], future studies should adapt a longitudinal design, which would in turn allow to draw more solid conclusions on causality and in turn pave the way for research on how students' Self-Control, NFC and AO could be fostered.

## Conclusion

By analyzing the interplay between NFC and AO in the prediction of Self-Control in 9th grade students, the present conceptual replication study strengthens theoretical assumptions that NFC and AO refer to cognitive and behavioral mechanisms that contribute to the successful exertion of Self-Control. It underlines the importance of taking interindividual differences into account in Self-Control research and findings from our mediation analysis indicate that Self-Control does not only depend on dispositions referring to behaviors closely associated with control processes (e.g., AO) but also on dispositions that go beyond the recruitment of control resources (e.g., NFC; more elaborated information processing, higher motivation to approach cognitive challenging situations). In light of the divergent finding in comparison to Grass et al [15] that the relation between NFC and Self-Control was found to be stronger for action-oriented students in secondary school whereas the relation was stable across different levels of AO in a more homogenous sample of university students, our moderation analysis highlights the importance of investigating relations between constructs identified for one age group (e.g., university) at other levels (e.g., secondary school) as they might not be fully generalizable across age groups but rather age-specific relations that might be specific to the

challenges of the respective (learning) environment. That observation is of considerable significance in the field psychology in which most studies tend to rely on higher education students [100,101]. From a developmental perspective, longitudinal studies from childhood and early adolescence on are needed to examine whether the relation between NFC and AO as well as processes underlying Self-Control change when children grow up.

## Supporting information

**S1 Appendix. Subgroup compared to full cohort.**
(DOCX)

**S2 Appendix. Descriptive statistics, parcel intercorrelations and covariance matrices.**
(DOCX)

## Acknowledgments

We would like to thank the national school monitoring team from the Luxembourg School Monitoring Programme for providing access to their database.

## Author Contributions

**Conceptualization:** Jonne Colling, Rachel Wollschläger, Julia Grass, Anja Strobel, Franzis Preckel, Antoine Fischbach.

**Formal analysis:** Jonne Colling, Ulrich Keller.

**Funding acquisition:** Antoine Fischbach.

**Investigation:** Rachel Wollschläger, Ulrich Keller.

**Methodology:** Jonne Colling, Ulrich Keller, Julia Grass.

**Supervision:** Franzis Preckel, Antoine Fischbach.

**Visualization:** Jonne Colling, Rachel Wollschläger.

**Writing – original draft:** Jonne Colling.

**Writing – review & editing:** Jonne Colling, Rachel Wollschläger, Julia Grass, Anja Strobel, Franzis Preckel, Antoine Fischbach.

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
