## [Decision Letter · Decision Letter 0]

21 Dec 2022

PONE-D-22-19185The relation between Self-Control, Need for Cognition and Action Orientation in secondary school students: A conceptual replication studyPLOS ONE

Dear Dr. Joanne Colling

Thank you for submitting your manuscript to PLOS ONE. After careful consideration, we feel that it has merit but does not fully meet PLOS ONE’s publication criteria as it currently stands. Therefore, we invite you to submit a revised version of the manuscript that addresses the points raised during the review process.

This conceptual replication study is an interesting paper on an important topic. However, it was done with secondary school students and was not mentioned what was the aim for using 9th graders clearly. As it is a quite long manuscript, especially the Introduction should be shortened at some points.  According to Reviewer 2; “a variable can be a mediator or a moderator.” I think this part should be described more clearly as a variable being both, mediator and moderator seems complicated and not understandable.

Reviewer 1

This study investigates the influence of need for cognition (NFC) and action orientation (AO) on self-control, focusing on the interplay of NFC and AO. The study is a conceptual replication from earlier research, focusing on 9th graders instead of university students. In line with the original study, the authors found that NFC and AO are correlated predictors of self- control. However, different to the original study they also found that AO moderates the relationship of NFC and AO (stronger influence for action-oriented participants).

The manuscript is well written and methodologically seemingly well done. I only have minor comments, as described in the following:

I did not completely understand the motivation behind the study. I fully agree that it is important to replicate earlier studies and investigate whether results generalize to different population groups. However, I would appreciate if you could provide more information why you specifically chose 9th graders (compared to university students). Is there any theoretical background, which suggests that there should be differences?In my opinion the abstract is quite long and should focus a bit more on the key information of the presented researchAt several points within the manuscript you provide information regarding the mean age of participants (e.g., l. 56). I would recommend to also add information on the standard deviation at this point, to give the reader a better impression how age was distributed within the sample.In the Introduction you refer to the delay of gratification study by Mischel et al. (1989) (e.g., l. 80) as well as to the ego-depletion theory by Baumeister (2002) (e.g., l. 210). Even though both studies are highly influential, there is also a lot of critique, especially concerning the replicability of these results. Please add respective information and references.I think the Introduction should be shortened at some points. For instance the definition of self-control is repeated at least four times within the manuscript.I would appreciate if you could add more information on the procedure of the EpStan (e.g., under which circumstances do the students fill out the questionnaire (at school vs at home)? In which order are questions presented?).As you describe within your data analysis section (l. 408), you replaced missing data with corresponding means. Even though this is an established practice to deal with missing data and I assume it is appropriate to do so in the given situation, there are also some problems with this method (e.g., it does not preserve relationships among variables and leads to an underestimation of standard errors). I do not expect you to use a different method, but I would appreciate if you could mention these problems in the manuscript, in order to make the reader aware of possible complications.

To sum up, after addressing the points mentioned above, the manuscript should be a good fit for Plos One. Good luck with your research!

Reviewer 2

First of all, I have to admit that I work in a quite different field, so when I read action and self I actually had other concepts in mind than the ones analyzed in this paper. So I think I am not familiar enough with the particular literature the authors are referring here to as to evaluate the knowledge that is gained (or not) in this paper. Maybe I should have declined this review. Anyway, I did it now.

To me the story seems sound. As do the methods and data. There are no real issues here. One might wonder whether different reliabilities of the scales are a problem (and a potential boundary for the correlations).

Yet, I have only one real issue where I struggled during reading. To my understanding, a variable can be a mediator or a moderator. Statistically it might the case that a clear moderation (i.e. an interaction) might lead to a correlation, that is if NFC and SC correlate but this correlation is moderated by AO wouldn’t than the mediation possibly be there, too, especially the partial one as found here? I am not entirely sure whether this is the case, but to me a variable being both, mediator and moderator seems, well, complicated at least. It gets even worse when it comes to theoretical grounds. Here, I admit, the authors lost me – it might be that I do not understand the statistics completely here, if so sorry, but at theoretical levels I can’t imagine how AO is moderator and mediator of the same NFC-SC correlation.

We look forward to receiving your revised manuscript.

Kind regards,

Ipek Gonullu, M.D., Ph.D.

Academic Editor

PLOS ONE

https://journals.plos.org/plosone/s/fileid=ba62/PLOSOne_formatting_sample_title_authors_affiliations.pdf.

2. For studies reporting research involving human participants, PLOS ONE requires authors to confirm that this specific study was reviewed and approved by an institutional review board (ethics committee) before the study began. Please provide the specific name of the ethics committee/IRB that approved your study, or explain why you did not seek approval in this case.

6. Please amend either the abstract on the online submission form (via Edit Submission) or the abstract in the manuscript so that they are identical.

Reviewers' comments:

Reviewer's Responses to Questions

**Comments to the Author**

1. Is the manuscript technically sound, and do the data support the conclusions?

Reviewer #1: Yes

Reviewer #2: Yes

2. Has the statistical analysis been performed appropriately and rigorously? 

Reviewer #1: Yes

Reviewer #2: Yes

3. Have the authors made all data underlying the findings in their manuscript fully available?

Reviewer #1: No

Reviewer #2: Yes

4. Is the manuscript presented in an intelligible fashion and written in standard English?

Reviewer #1: Yes

Reviewer #2: Yes

5. Review Comments to the Author

Reviewer #1: Comments on PONE-D-22-19185:

This study investigates the influence of need for cognition (NFC) and action orientation (AO) on self-control, focusing on the interplay of NFC and AO. The study is a conceptual replication from earlier research, focusing on 9th graders instead of university students. In line with the original study, the authors found that NFC and AO are correlated predictors of self-control. However, different to the original study they also found that AO moderates the relationship of NFC and AO (stronger influence for action-oriented participants).

The manuscript is well written and methodologically seemingly well done. I only have minor comments, as described in the following:

• I did not completely understand the motivation behind the study. I fully agree that it is important to replicate earlier studies and investigate whether results generalize to different population groups. However, I would appreciate if you could provide more information why you specifically chose 9th graders (compared to university students). Is there any theoretical background, which suggests that there should be differences?

• In my opinion the abstract is quite long and should focus a bit more on the key information of the presented research

• At several points within the manuscript you provide information regarding the mean age of participants (e.g., l. 56). I would recommend to also add information on the standard deviation at this point, to give the reader a better impression how age was distributed within the sample.

• In the Introduction you refer to the delay of gratification study by Mischel et al. (1989) (e.g., l. 80) as well as to the ego-depletion theory by Baumeister (2002) (e.g., l. 210). Even though both studies are highly influential, there is also a lot of critique, especially concerning the replicability of these results. Please add respective information and references.

• I think the Introduction should be shortened at some points. For instance the definition of self-control is repeated at least four times within the manuscript.

• I would appreciate if you could add more information on the procedure of the EpStan (e.g., under which circumstances do the students fill out the questionnaire (at school vs at home)? In which order are questions presented?).

• As you describe within your data analysis section (l. 408), you replaced missing data with corresponding means. Even though this is an established practice to deal with missing data and I assume it is appropriate to do so in the given situation, there are also some problems with this method (e.g., it does not preserve relationships among variables and leads to an underestimation of standard errors). I do not expect you to use a different method, but I would appreciate if you could mention these problems in the manuscript, in order to make the reader aware of possible complications.

To sum up, after addressing the points mentioned above, the manuscript should be a good fit for Plos One. Good luck with your research!

Reviewer #2: First of all, I have to admit that I work in a quite different field, so when I read action and self I actually had other concepts in mind than the ones analyzed in this paper. So I think I am not familiar enough with the particular literature the authors are referring here to as to evaluate the knowledge that is gained (or not) in this paper. Maybe I should have declined this review. Anyway, I did it now.

To me the story seems sound. As do the methods and data. There are no real issues here. One might wonder whether different reliabilities of the scales are a problem (and a potential boundary for the correlations).

Yet, I have only one real issue where I struggled during reading. To my understanding, a variable can be a mediator or a moderator. Statistically it might the case that a clear moderation (i.e. an interaction) might lead to a correlation, that is if NFC and SC correlate but this correlation is moderated by AO wouldn’t than the mediation possibly be there, too, especially the partial one as found here? I am not entirely sure whether this is the case, but to me a variable being both, mediator and moderator seems, well, complicated at least. It gets even worse when it comes to theoretical grounds. Here, I admit, the authors lost me – it might be that I do not understand the statistics completely here, if so sorry, but at theoretical levels I can’t imagine how AO is moderator and mediator of the same NFC-SC correlation.

6. PLOS authors have the option to publish the peer review history of their article (what does this mean?). If published, this will include your full peer review and any attached files.

Reviewer #1: No

Reviewer #2: No

---

## [Author Response · Author response to Decision Letter 0]

18 Feb 2023

Revision of manuscript PONE-D-22-19185

Title: The relation between Self-Control, Need for Cognition and Action Orientation in secondary school students: A conceptual replication study

Reply to Editor: 

Thank you for submitting your manuscript to PLOS ONE. After careful consideration, we feel that it has merit but does not fully meet PLOS ONE’s publication criteria as it currently stands. Therefore, we invite you to submit a revised version of the manuscript that addresses the points raised during the review process. 

Before addressing the mentioned concerns, we would like to thank the Editor for considering our submission to PLOS ONE and for the time and effort in commenting our manuscript. We were pleased to read that the reviewers found merit in our work, and we appreciated the constructive comments and suggestions to improve our contribution to the field. We did our best to integrate the feedback meticulously into the manuscript and hope that it now meets the publication criteria. 

This conceptual replication study is an interesting paper on an important topic. However, it was done with secondary school students and was not mentioned what was the aim for using 9th graders clearly. 

We would like to thank the Editor and Reviewer #1 for pointing out that more information on why we chose 9th graders is needed to fully understand the motivation behind our study. To address this comment in the manuscript, we added a paragraph to the section “The present study” on page 15, in which we present research findings that have identified adolescence as a pivotal period for the development of self-regulatory capacities due to adolescent-specific neurocognitive processes, and important cognitive and social challenges that result in higher and new demands on adolescents’ self-regulatory capacities (e.g., learning environment of secondary school characterized by more challenging academic structures and expectations, greater individual responsibility in managing academic requirements). Against this background, we argue that it seems important to examine the interplay between Self-Control, NFC and AO in such a relevant period for self-regulation and the approach of a conceptual replication study furthermore allows generate a first understanding of whether relations between constructs reported at one level of school (e.g., university) are already established in a younger age group or whether age-specific relations can be identified that might be specific to the challenges of the respective learning environment (e.g., secondary education). 

As it is a quite long manuscript, especially the Introduction should be shortened at some points. 

As described in more detail as a reply to a comment made by Reviewer #1, we have shortened our manuscript by including definitions only at one point within the manuscript, by excluding detailed information that does not seem strictly necessary to understand the present study’s aim and through a reduction of examples given for a certain aspect. All changes made to shorten the introduction are highlighted in the manuscript. In total, the introduction was shortened by 484 words compared to the original submission.

According to Reviewer 2; “a variable can be a mediator or a moderator.” I think this part should be described more clearly as a variable being both, mediator and moderator seems complicated and not understandable.

Thank you for pointing out the difficulties that are coming with having one variable being both a mediator and a moderator of the same relation between two other variables of interest. As explained in more detail in our response to the comments made by Reviewer #2, we consider the knowledge on the interplay between Self-Control, NFC and AO still to scarce to favor one of the two hypotheses (mediation vs. moderation) over the other and have therefore examined in line with the original study both hypotheses. To make our theoretical assumptions easier to follow, we have reformulated our hypotheses to allow a more encompassing understanding of the theoretical assumptions standing behind the consideration of AO as both mediator and moderator (see (1) and (3) further below in our response to Reviewer #2).

On the statistical level, we have been aiming at facilitating the understanding of the complex matter of considering the same variable as both mediator and moderator of a relation by adding clearer definitions of what moderator and mediator variable are. In addition, we referred to an article by Judd et al. (2001) that gives an encompassing explanation of how one variable can be considered both as mediator and moderator of the same relation between two other variables. This example is also summarized in detail in our response to Reviewer #2. To guide the reader through the theoretical embedding of our findings in the “Discussion” section, we furthermore added two sentences on page 31 and 32 respectively, which are referring back to the definitions of mediator and moderator variables to allow a better understanding of the conclusions drawn from the respective analyses. 

Revision of manuscript PONE-D-22-19185

Title: The relation between Self-Control, Need for Cognition and Action Orientation in secondary school students: A conceptual replication study

Reply to Reviewer #1: 

This study investigates the influence of need for cognition (NFC) and action orientation (AO) on self-control, focusing on the interplay of NFC and AO. The study is a conceptual replication from earlier research, focusing on 9th graders instead of university students. In line with the original study, the authors found that NFC and AO are correlated predictors of self- control. However, different to the original study they also found that AO moderates the relationship of NFC and AO (stronger influence for action-oriented participants). The manuscript is well written and methodologically seemingly well done. I only have minor comments, as described in the following:

We would like to thank Reviewer #1 for the time and effort in reviewing our manuscript and for the helpful comments. 

(1) I did not completely understand the motivation behind the study. I fully agree that it is important to replicate earlier studies and investigate whether results generalize to different population groups. However, I would appreciate if you could provide more information why you specifically chose 9th graders (compared to university students). Is there any theoretical background, which suggests that there should be differences?

We would like to thank Reviewer #1 for pointing out that more information on why we chose 9th graders is needed to understand the motivation behind our study. To address this comment in the manuscript, we added the following paragraphs to the section “The present study” on page 15: 

In research, adolescence has been identified as a pivotal period for the development of self-regulatory capacities due to adolescent-specific neurocognitive processes (e.g., brain maturation, [63]). In addition to these biological processes, adolescences is a developmental period characterized by important cognitive and social challenges [64,65] that result in new demands on self-regulatory capacities. One of these challenges consists in the transition into secondary school, which can be considered a learning environment of special interest as it is characterized by more challenging academic structures and expectations [66]. Considering that students are faced with a higher workload [67], with an emphasis on performance goals and evaluations of achievement in competition to others [68,69], and with greater individual responsibility in managing academic requirements [66], a greater emphasis is put on secondary school students (academic) self-regulation capacities [70]. 

Against this background, it seems important to examine motivational constructs such as Self-Control and NFC (including their interplay with other constructs such as AO) in such a relevant period for the development of self-regulatory capacities. The present conceptual replication study with its focus on 9th graders allows to generate a first understanding of whether relations between constructs reported at one level of school (e.g., university) are already established in a younger age group or whether age-specific relations can be identified that might be specific to the challenges of the respective learning environment (e.g., secondary education).

(2) In my opinion the abstract is quite long and should focus a bit more on the key information of the presented research. 

We followed your recommendation and shortened the first part of the abstract by 94 words. We excluded non-necessary information and reduced definitions of the constructs investigated. The manuscript highlights these changes and the abstract now reads as follows: 

Self-Control can be defined as the self-initiated effortful process that enables individuals to resist temptation impulses. It is relevant for conducting a healthy and successful life. For university students, Grass et al. [1] found that Need for Cognition as the tendency to engage in and enjoy thinking, and Action Orientation as the flexible recruitment of control resources in cognitively demanding situations, predict Self-Control. Further, Action Orientation partially mediated the relation between Need for Cognition and Self-Control. In the present conceptual replication study, we investigated the relations between Self-Control, Need for Cognition, and Action Orientation in adolescence (N = 892 9th graders) as a pivotal period for the development of self-control. We replicated the findings that Need for Cognition and Action Orientation predict Self-Control and that Action Orientation partially mediates the relation between Need for Cognition and Self-Control. In addition, we found that Action Orientation moderates the relation between Need for Cognition and Self-Control. This result implies that in more action-oriented students Need for Cognition more strongly predicted Self-Control than in less action-oriented students. Our findings strengthen theoretical assumptions that Need for Cognition and Action Orientation are important cognitive and behavioral mechanisms that contribute to the successful exertion of Self-Control. 

(3) At several points within the manuscript, you provide information regarding the mean age of participants (e.g., l. 56). I would recommend to also add information on the standard deviation at this point, to give the reader a better impression how age was distributed within the sample.

Thank you for pointing this out. We added the standard deviation each time the mean age of participants was indicated in the original study (e.g., l. 94, l. 105, and l. 181). 

(4) In the Introduction you refer to the delay of gratification study by Mischel et al. (1989) (e.g., l. 80) as well as to the ego-depletion theory by Baumeister (2002) (e.g., l. 210). Even though both studies are highly influential, there is also a lot of critique, especially concerning the replicability of these results. Please add respective information and references.

We agree with this suggestion and added information and references on the critical reception of Mischel et al.’s (1989) delay of gratification study and Baumeister’s (2002) model of ego-depletion. The following sentences have been included in the section “Self-Control” on page 6 (1) and “NFC and Action Orientation as predictors of Self-Control” on page 14 (2), respectively: 

(1) Although the paradigm has been criticized with regard to its replicability (e.g., smaller and rarely statistically significant associations) and the Marshmallow Test measures not only self-control but also other essential aspects (e.g., [20]), the findings nevertheless illustrate the positive effects of self-regulatory abilities.

(2) Whereas assumptions of Baumeister’s [49] model of ego-depletion were supported by a meta-analysis [58], recent studies critically discuss the existence of ego-depletion (e.g., [59]) and furthermore highlight the challenges that are related to research on a multicomponent phenomenon (e.g., selection of appropriate ego-depletion tasks, lack of clear operational definitions of Self-Control; [60–62]).

(5) I think the Introduction should be shortened at some points. For instance, the definition of self-control is repeated at least four times within the manuscript.

We shortened the introduction by including definitions only at one point within the manuscript with the exception of line 208 to 211, where parts of the Self-Control definition are repeated to allow a meaningful introduction of the example of a student who needs to choose between studying for a math test or checking social media. In addition, we excluded other repetitions such as a sentence that NFC “is associated not only with central information processing and the tendency to approach cognitive challenges but also with the actual investment of cognitive effort in order to successfully implement intentions to exert Self-Control” (see line 58 in the section “Introduction” and line 241 in the section “NFC and Action Orientation as predictors as Self-Control”). To further shorten the introduction, we scanned the manuscript for information that is not necessary to understand the aim of our study (e.g., the information that Self-Control is considered as a type or prerequisite of Self-Regulation in line 72 in the section “Self-Control”) and reduced examples given for a certain aspect. All changes are highlighted in the manuscript and in total, the introduction was shortened by 484 words compared to the original submission.

(6) I would appreciate if you could add more information on the procedure of the EpStan (e.g., under which circumstances do the students fill out the questionnaire (at school vs at home)? In which order are questions presented?).

In order to clarify the procedure of the ÉpStan, we have added additional sentences to the section “Procedure and participants” on page 19 (1) and we added details on the order in which the items were presented to the section “Measures” on page 20 (2): 

(1) The questionnaire is presented computer-based. Students completed it at school in the presence of a teacher after having taken the academic achievement tests. All answers were confidential and directly stored on a dedicated platform without requiring a manual encoding.

The student questionnaire including the measures that are of relevance for the present study was presented to a subsample of N = 1.678 students with the NFC scale presented first, followed by the Self-Control scales and the AO scale, respectively.

(2) Items of both subscales were presented alternately.

(7) As you describe within your data analysis section (l. 408), you replaced missing data with corresponding means. Even though this is an established practice to deal with missing data and I assume it is appropriate to do so in the given situation, there are also some problems with this method (e.g., it does not preserve relationships among variables and leads to an underestimation of standard errors). I do not expect you to use a different method, but I would appreciate if you could mention these problems in the manuscript, in order to make the reader aware of possible complications.

Thank you for pointing this out. To make the reader aware of possible shortcomings of this handling of missings, we have added the following sentence to the section “Data analysis” on page 22: 

For students with no answers on up to 30 % of the items of each scale, the mean score of the answered items on the total scale replaced missing values before sum scores were created for each measure. Although replacing missing values by the mean score of the answered items on the total scale is an established method when it comes to handling missing data (e.g., [82]), it has some important shortcomings such as the underestimation of standard errors, the loss of information on the relationship between variables, and the artificial deflation of a variable’s variance that need to be taken into consideration when interpreting the results [83–85]. Nevertheless, this procedure respects the manual guidelines for treating missing data in the NFC-KIDS scale [39] and as no clear indications were found for the other measures of the present study, the 70 % rule was subsequently applied to the scales of Self-Control and AO.

To sum up, after addressing the points mentioned above, the manuscript should be a good fit for PLOS One. Good luck with your research!

Thank you for your valuable input and for the opportunity to implement your feedback in a revised version of our manuscript.  

Reply to Reviewer #2: 

First of all, I have to admit that I work in a quite different field, so when I read action and self I actually had other concepts in mind than the ones analyzed in this paper. So I think I am not familiar enough with the particular literature the authors are referring here to as to evaluate the knowledge that is gained (or not) in this paper. Maybe I should have declined this review. Anyway, I did it now. To me the story seems sound. As do the methods and data. There are no real issues here.

We would like to thank Reviewer #2 for conducting this review and for the helpful comments, especially for pointing out the complexity of considering AO as mediator and moderator of the relation between NFC and Self-Control. In the following, we are going to address the concerns raised by Reviewer #2: 

One might wonder whether different reliabilities of the scales are a problem (and a potential boundary for the correlations).

Thank you for pointing this out. Although the different reliabilities might affect the correlations between the manifest item parcels in Table A5 in S2 Appendix B, the usage of Structural Equation Modeling to explore the main research questions allows accounting for measurement errors when exploring complex relations between latent variables (e.g., Newsom, 2015). The finding of NFC and AO being correlated predictors of Self-Control should therefore not be affected by the different scale reliabilities. 

Yet, I have only one real issue where I struggled during reading. To my understanding, a variable can be a mediator or a moderator. Statistically it might the case that a clear moderation (i.e. an interaction) might lead to a correlation, that is if NFC and SC correlate but this correlation is moderated by AO wouldn’t than the mediation possibly be there, too, especially the partial one as found here? I am not entirely sure whether this is the case, but to me a variable being both, mediator and moderator seems, well, complicated at least. It gets even worse when it comes to theoretical grounds. Here, I admit, the authors lost me – it might be that I do not understand the statistics completely here, if so sorry, but at theoretical levels I can’t imagine how AO is moderator and mediator of the same NFC-SC correlation.

Although considering one variable (e.g., AO) as both mediator and moderator of a relation between two other variables (e.g., NFC and Self-Control) is rather complex on the statistical and theoretical level, there are theoretical assumptions standing behind the consideration of AO as both mediator and moderator that underline the importance of investigating whether AO (partially) mediates and moderates the relation between NFC. In line with the original study by Grass et al., it can be assumed that lower levels of AO might result in less recruitment of cognitive resources and a reduced motivation to invest cognitive effort even in individuals high in NFC (i.e., weaker relation between NFC and Self-Control). Higher levels of AO, on the other hand, could foster the actual investment of cognitive effort (i.e., stronger relation between NFC and Self-Control). Additionally, it can be expected that AO (partially mediates) the relation between NFC and Self-Control as we assume additional processes linking NFC to Self-Control that go beyond the recruitment of control resources (e.g., elaborated information processing, higher motivation to approach cognitively challenging situations). Due to the still scarce knowledge on the interplay between Self-Control, NFC and AO, it does not seem possible to favor one of the two hypotheses over the other and we have therefore examined both hypotheses analogously to the original study. To make our theoretical assumptions easier to follow, we have reformulated our hypotheses to allow a more encompassing understanding of the theoretical assumptions standing behind the consideration of AO as both mediator and moderator (see (1) and (3) further below.

On the statistical level, it is also possible that one variable can be both a moderator and a mediator as described in Judd et al. (2001) for the example of study time. The authors introduce a study design in which performance differences are identified between a treatment group of students that has been taught with a new curriculum while a control group of students has been following the traditional one. In this context, the authors argue that the new curriculum might increase the students’ interest in the subject, convey the material more clearly or encourage the students to study harder outside of their classes – mediators that might in turn result in a higher performance in the treatment group. Besides potential mediators, factors such as classroom size or assigned teachers might affect the magnitude of the performance difference and have thus to be considered as a moderator of the treatment effect. 

Judd et al. (2001, p.115) refer to the example of study time stating that “it is possible that the same variable may serve as both a mediator and a moderator.” Coming back to the example introduced above, the new curriculum might result in a higher performance as it causes students to study more (study time as mediator), and study time might furthermore result in a stronger relation between the new curriculum and academic performance for those students that spend more time studying (study time as moderator). 

 Following this line of thought with regard to our investigation, it can be expected that AO partially mediates the relation between NFC and Self-Control as we assume additional processes linking NFC to Self-Control that go beyond the recruitment of control resources (e.g., elaborated information processing, higher motivation to approach cognitively challenging situations). Besides mediating the relation between NFC and Self-Control, AO can in addition be assumed to moderate the relation by affecting its strength (e.g., stronger relation for students high in AO). 

 To facilitate the reader’s understanding, we added in a first step clear definitions of a moderator (1) and mediator variable (2) to “The present study” on page 17 to 18 of our manuscript. In addition, we referred to the paper by Judd et al. (2001) in order to guide the reader to an encompassing explanation of how one variable can be considered both as mediator and moderator of the same relation between two other variables (3). To guide the reader through the theoretical embedding of our findings in the “Discussion” section on page 31 and 32, we furthermore added two sentences that refer back to the definitions of mediator and moderator variables to allow for a better understanding of the conclusions drawn from the respective analyses (4). In addition, we adapted two sentences in the section “Conclusion” on page 38 by linking our findings to theoretical argumentations on potential mediation/moderation effects presented beforehand (5). 

(1) We therefore investigated whether AO moderates the relation between NFC and Self-Control. A moderator variable affects the strength of the relation between two other variables (e.g., NFC and Self-Control) in such a way that this strength differs depending on the level of the moderator (e.g., AO, [73,74]). Applied to the constructs of interest in the present study, lower levels of AO might result in less recruitment of cognitive resources and a reduced motivation to invest cognitive effort even in individuals high in NFC (i.e., weaker relation between NFC and Self-Control). Higher levels of AO, on the other hand, could foster the actual investment of cognitive effort (i.e., stronger relation between NFC and Self-Control).

(2) A mediator variable is defined as the intermediary process or mechanism through which one variable relates to another variable, or in other words, it allows to understand the relation more completely by assessing the extent to which the relation between two variables (e.g., NFC and Self-Control) is direct or indirect via a mediator (e.g., AO, [73,74,76]). 

(3) Besides having an impact on the strength of the relation between two variables (moderation), the same variable can also mediate the relation between the two variables in question (e.g., [74], see [75] for a detailed explanatory example). Following this line of thought with regard to our investigation, it can thus be expected that AO partially mediates the relation between NFC and Self-Control besides moderating it as we assume additional processes linking NFC to Self-Control that go beyond the recruitment of control resources (e.g., elaborated information processing, higher motivation to approach cognitively challenging situations; see Fig 1 in [1] for a visualization).

(4) A mediation analysis allows to understand the relation between two variables more completely by assessing the extent to which the relation is affected by an intermediary process – the mediator variable [73,74,76].

Besides mediating the relation between two variables, the same variable can also moderate the relation between the two variables in question and affect its strength depending on the level of the moderator variable [74–76].

(5) It underlines the importance of taking interindividual differences into account in Self-Control research and findings from our mediation analysis indicate that Self-Control does not only depend on dispositions referring to behaviors closely associated with control processes (e.g., AO) but also on dispositions that go beyond the recruitment of control resources (e.g., NFC; more elaborated information processing, higher motivation to approach cognitive challenging situations).

In light of the divergent finding in comparison to Grass et al [1] that the relation between NFC and Self-Control was found to be stronger for action-oriented students in secondary school whereas the relation was stable across different levels of AO in a more homogenous sample of university students, our moderation analysis highlights the importance of investigating relations between constructs identified for one age group (e.g., university) at other levels (e.g., secondary school) as they might not be fully generalizable across age groups but rather age-specific relations that might be specific to the challenges of the respective (learning) environment.

Thank you for your valuable input and for the opportunity to implement your feedback in a revised version of our manuscript.

---

## [Decision Letter · Decision Letter 1]

23 May 2023

The relation between Self-Control, Need for Cognition and Action Orientation in secondary school students: A conceptual replication study

PONE-D-22-19185R1

Dear Dr. Colling,

We’re pleased to inform you that your manuscript has been judged scientifically suitable for publication and will be formally accepted for publication once it meets all outstanding technical requirements.

Kind regards,

Ipek Gonullu, M.D., Ph.D.

Academic Editor

PLOS ONE

Additional Editor Comments (optional):

Reviewers' comments:

Reviewer's Responses to Questions

**Comments to the Author**

1. If the authors have adequately addressed your comments raised in a previous round of review and you feel that this manuscript is now acceptable for publication, you may indicate that here to bypass the “Comments to the Author” section, enter your conflict of interest statement in the “Confidential to Editor” section, and submit your "Accept" recommendation.

Reviewer #1: All comments have been addressed

2. Is the manuscript technically sound, and do the data support the conclusions?

Reviewer #1: Yes

3. Has the statistical analysis been performed appropriately and rigorously? 

Reviewer #1: Yes

4. Have the authors made all data underlying the findings in their manuscript fully available?

Reviewer #1: No

5. Is the manuscript presented in an intelligible fashion and written in standard English?

Reviewer #1: Yes

6. Review Comments to the Author

Reviewer #1: The authors put a lot of effort in revising their manuscript, which I highly appreciate. All my comments have been addressed accordingly, thank you! The paper now is much better readable but also includes all relevant information for the reader. This is very nice.

Therefore, I am happy to recommend publication of this interesting article. I just noticed two very minor issues, which the authors might want to address before publication:

• In line 14 of the Abstract, you should either remove the dot after “Self-Control” or the “and” after the dot

• In line 593 you state: “RMSEA = .000 (90 % Confidence Interval: [.000; .009], p = 1.000) “. Just like a p-value of “.000” also a value of “1.000” rarely is the correct description for a statistical analysis. I would recommend to put “>.999” instead and the RMSEA should be indicated as “<.001”.

7. PLOS authors have the option to publish the peer review history of their article (what does this mean?). If published, this will include your full peer review and any attached files.

Reviewer #1: **Yes: **Moritz Reis

---

## [Editor Report · Acceptance letter]

1 Jun 2023

PONE-D-22-19185R1 

The relation between Self-Control, Need for Cognition and Action Orientation in secondary school students: A conceptual replication study 

Dear Dr. Colling:

I'm pleased to inform you that your manuscript has been deemed suitable for publication in PLOS ONE. Congratulations! Your manuscript is now with our production department. 

Kind regards, 

on behalf of

Asistant Professor Ipek Gonullu 

Academic Editor

PLOS ONE